# Nutritional Factors: Benefits in Glaucoma and Ophthalmologic Pathologies

**DOI:** 10.3390/life13051120

**Published:** 2023-04-30

**Authors:** Mutali Musa, Marco Zeppieri, George Nnamdi Atuanya, Ehimare S. Enaholo, Efioshiomoshi Kings Topah, Oluwasola Michael Ojo, Carlo Salati

**Affiliations:** 1Department of Optometry, University of Benin, Benin City 300238, Edo State, Nigeria; 2Department of Ophthalmology, University Hospital of Udine, 33100 Udine, Italy; 3Centre for Sight Africa, Nkpor, Onitsha 434109, Anambra State, Nigeria; 4Department of Optometry, Faculty of Allied Health Sciences, College of Health Sciences Bayero University, Kano 700006, Kano State, Nigeria; 5School of Optometry and Vision Sciences, College of Health Sciences, University of Ilorin, Ilorin 240003, Kwara State, Nigeria

**Keywords:** nutrient, glaucoma, ophthalmology, glutathione, minocycline, fisetin, omega-3, zeaxanthin, lutein, resveratrol

## Abstract

Glaucoma is a chronic optic neuropathy that can lead to irreversible functional and morphological damage if left untreated. The gold standard therapeutic approaches in managing patients with glaucoma and limiting progression include local drops, laser, and/or surgery, which are all geared at reducing intraocular pressure (IOP). Nutrients, antioxidants, vitamins, organic compounds, and micronutrients have been gaining increasing interest in the past decade as integrative IOP-independent strategies to delay or halt glaucomatous retinal ganglion cell degeneration. In our minireview, we examine the various nutrients and compounds proposed in the current literature for the management of ophthalmology diseases, especially for glaucoma. With respect to each substance considered, this minireview reports the molecular and biological characteristics, neuroprotective activities, antioxidant properties, beneficial mechanisms, and clinical studies published in the past decade in the field of general medicine. This study highlights the potential benefits of these substances in glaucoma and other ophthalmologic pathologies. Nutritional supplementation can thus be useful as integrative IOP-independent strategies in the management of glaucoma and in other ophthalmologic pathologies. Large multicenter clinical trials based on functional and morphologic data collected over long follow-up periods in patients with IOP-independent treatments can pave the way for alternative and/or coadjutant therapeutic options in the management of glaucoma and other ocular pathologies.

## 1. Introduction

Glaucoma is an ocular disease that is characterized by the progressive loss of optic nerve fibers, which results in visual field defects. Glaucoma sits at the top among the causes of irreversible blindness worldwide [1]. Predisposing factors include heredity, gender, race, trauma, and steroid use [2]. Obstructive sleep apnea has also been implicated in glaucoma [3]. Glaucoma can either be primary or secondary [4,5,6]. Glaucoma is diagnosed using tests such as tonometry [7,8], visual field measurement, and direct observation. Glaucoma morbidity costs USD 3 billion annually in the USA alone [9]. Varma et al. projected glaucoma to have a worldwide prevalence of above 79 million by 2020 [10].

Topical drugs, lasers, and surgery are used to manage glaucoma [5]. The only modifiable factor that is currently clinically considered when treating glaucoma to avoid progression is intraocular pressure (IOP) [7]. IOP is the most important risk factor in the development and worsening of this disease. The diagnosis of glaucoma is typically based on IOP in the presence of optic nerve fiber layer thinning, glaucomatous optic nerve cupping, and visual field loss [8].

Nutritional supplements have been proposed in the past years to offer neuroprotection and anti-oxidative factors to help slow down glaucomatous progressive loss. Nutrients have been used in medicine in the form of supplements for the proper function of body systems [11]. Most nutritional elements are typically found in a healthy diet [12,13] and are favored in the presence of non-dietary sources, including the environment [14]. This approach is now being termed as “Nutraceuticals” [15]. Natural immunity largely depends on the availability of a nutrient-rich diet [16,17]. A complex relationship exists between nutrients and infectious processes [18,19,20]. The knowledge of nutrients in ophthalmic medicine is limited, but the current literature seems to be expanding in this interesting field [21]. Nutritional factors have been implicated in multiple oculo-visual diseases [22,23,24]. A thorough understanding of the molecular mechanisms and pathways involved in nutrients can be useful for considering this supplementary therapy in a routine clinical setting when managing glaucoma and other ophthalmologic conditions.

It is well-known and undebatable that the IOP is the most important factor for this disease; however, glaucoma is not all about pressure, as previously thought. The aim of this minireview is to examine the molecular and biological characteristics of some of these nutrients that have shown potential benefits for ophthalmologic pathologies, especially for glaucoma. For each nutrient considered, we have provided: a brief summary of the molecular and chemical composition; the properties, biochemical pathways, and mechanism of action; the beneficial uses for medicine reported in the literature; and the potential advantages and limitations of the nutrients when considered in ophthalmologic disorders, with particular focus on glaucoma. Thus, we apologize in advance if opinion leaders and experts in this field of study have not been cited in our paper.

## 2. Materials and Methods

Our study assessed some of the current literature regarding the use of several nutrients in medicine and modern-day ophthalmology.

A search of the PubMed database was carried out using the query below: “(“dietary supplements” [MeSH terms] OR (“dietary” [all fields] AND “supplements” [all fields]) OR “dietary supplements” [all fields] OR “supplement” [all fields] OR “supplement s” [all fields] OR “supplemented” [all fields] OR “supplementing” [all fields] OR “supplements” [all fields]) AND (“eye” [MeSH terms] OR “eye” [all fields])”.

The inclusion criteria were: studies on selected nutrients; studies with a focus on ophthalmology; full text articles that were available on PubMed; journal articles published in the last decade (2012–2023); publications that were originally published in the English language; and primary sources with qualitative or quantitative research designs. The exclusion criteria included: studies not published in English; duplicate studies; and studies outside of the scope of this work. Each study was independently assessed by at least two reviewers (M.M., G.A., E.E., E.T., and M.Z), and rating decisions were based on the consensus of the reviewing authors. A total of 374 references were included in the review, as indicated in the PRISMA diagram in Figure 1. This search strategy was a limitation employed in light of the vast amount of literature available, which could have thus potentially and unintentionally excluded opinion leaders in this field of research.

The Preferred Reporting Items for Systematic Reviews and Meta-Analyses (PRISMA) [25] was used to populate this review using the PubMed database in February 2023, which is shown in Figure 1.

Table 1 lists clinical studies that have reported the benefits of specific nutrients in glaucoma and other ophthalmologic pathologies.

## 3. Results

### 3.1. Glutathione

Glutathione is an organosulfur molecule that derives from the family of thiols. It is a very potent antioxidant that can be found all over the body. It is a tri-peptide comprising the amino acids cysteine, glutamate, and glutamic acid and has similar concentrations to glucose and cholesterol. The glutathione molecule is produced within the cytoplasm. It is present in the body in either an initial reduced or oxidized state; the reduced form is about 100 times more abundant than the oxidized form in resting cells. This abundance drops 10-fold in the presence of oxidative stress [42]. This ratio is referred to as the redox number of the cell.

Glutathione generally prevents damage from oxidative stress that is triggered by free radicals, oxides, and toxic xenobiotics [43]. It facilitates the replenishment of depleted ascorbic acid, tocopherols, and tocotrienols. This substance catalyzes the outward movement of mercury out of neurons and regulates cell apoptosis.

Glutathione levels have been positively correlated with physical health and well-being [44]; it is especially important for people with fatty liver diseases. Glutathione has been implicated in the induction of ferroptosis in thyroid cancer, increase in cancerous cell death, reduction in the migration ability of the diseased cells, and shortened hospital durations [45]. Glutathione has also been reviewed to address some of the clinical symptoms associated with COVID-19 [46]. In the field of dermatology, this substance has been shown to have both antiaging and anti-melanogenic properties [47].

Glutathione exhibits a potent antioxidant function in the eye because it has a high affinity for reactive oxygen molecules. It is also a cofactor of glutathione transferases and peroxidases [48]. Takeshi et al. [27] investigated glutathione concentrations as a marker for mitochondrial defects in glaucoma patients. The results of this study showed significantly elevated glutathione disulfide in glaucoma subjects compared with the controls with a smaller redox index. This lower redox index led them to conclude that mitochondrial defects likely result in vascular changes that precipitate glaucomatous damage. Gherghel et al. compared blood work for POAG and NTG patients and compared them with normal patients. The authors found that significantly lower levels of reduced glutathione levels were significantly lower than the levels in the non-glaucomatous controls [28].

### 3.2. Minocycline

Minocycline is a second-generation tetracycline-class antibiotic. It is used for both infective and non-infective conditions and also has apoptotic applications. It was created in 1967 as a semi-synthetic broad-spectrum antibiotic. Minocycline works by binding to the 30s ribosome of organisms, stopping such organisms from replicating or growing. This substance is thus bacteriostatic. Minocycline is more easily absorbed into the skin and the central nervous system than other tetracyclines. It has also been shown to have anti-inflammatory properties [49].

Minocycline binds to the 30s ribosome receptors in the cells of prokaryotes, stopping the activities of transfer RNA molecules that help with protein synthesis. This halts cell growth and results in a bacteriostatic effect. Minocycline is very fat-soluble and is therefore easily absorbed all over the human body. It can be administered orally and intravenously. Minocycline is useful for managing many antibacterial-resistant infections [50,51].

Minocycline has primarily been used for decades for its effects on both Gram-positive and Gram-negative bacterial action [50]. Studies have reported that it can induce neuroprotection in rats that are undergoing temporary middle cerebral artery occlusion [52]. IV minocycline has also been found to be potent against multiple multi-resistant drug organisms [51]. Garner et al. reported the benefits of minocycline for acne vulgaris [53].

Using a rat model, minocycline has been reported to upregulate pro-survival genes in glaucoma [29]. The neuroprotective effects seem to be activated by the anti-apoptosis (Bcl-2 gene expression) and a drop in IL-8 gene expression in the retina. It has also been found to reduce activated microglia levels in rabbit eyes. Grotegut et al. [54] induced optic nerve (ON) degeneration in rats using an intravitreal injection of S100B; this substance is a calcium-binding protein that is known to induce characteristic glaucomatous damage when administered to living specimens. The authors further reported that 13.5 mg/kg of minocycline was able to inhibit microglia expression and therefore protect the neurofilaments of the ON after an intravenous administration of S100B [55,56].

### 3.3. Spermidine 

Spermidine is a naturally occurring polyamine in the cells of organisms. They are known to mediate anti-aging processes but tend to be limited by the increasing age of the organism [57]. Polyamines are commonly occurring, positively charged compounds. It was originally derived from semen, hence the origin of this name. The chemical formula is C7H19N3. Spermidine is derived from putrescine in a reaction catalyzed by spermidine synthase. Dietary sources include proteins such as soy, legumes, and grains.

Increased dietary supplementation with spermidine has been shown to improve overall health [58]. Spermidine blocks the enzyme that catalyzes nitric oxide synthesis, aids DNA formation, and regulates the growth of other polyamines such as spermine and thermospermine. The official IUPAC name of this substance is N-(3-aminopropyl)butane-1,4-diamine. The antiaging abilities of spermidine are mediated by multiple pathways, including autophagy [59], hypusination [60], and the stabilization of melanogenesis [61].

Modeo et al. have postulated that spermidine mediates macrophagy and autophagic processes. They further suggested that dietary augmentation with spermidine can be correlated with reduced cardiovascular pathology [61]. Lefèvre et al. suggested that polyamines generally function in enhancing sperm motility and the development of ovarian follicles in mammalian models [62]. The role of spermidine has been highlighted in the preservation of mitochondrial function, protein deacetylation, and anti-inflammation [63]. Yousefi-Manesh et al. demonstrated that spermidine reduced neuropathic pain in injury-induced neuropathy in a murine model [64]. Spermidine has also been shown to help maintain hormonal balance and generally reduce oxidative stress [64].

The potential benefits of spermidine in ocular diseases are far-reaching [65]. The defective mitochondrial oxidation of substrates, which is mediated by spermidine, is one of the pathophysiology channels that is seen in glaucoma [66]. Buisset et al. [67] reported a reduction in spermine levels in the eyes of patients with open-angle glaucoma. Spermine is a derivative of spermidine. Other studies have equally reported reduced spermidine levels in glaucomatous eyes [66]. Lillo et al. reported, however, elevated levels of spermidine in the aqueous humor of open-angle glaucoma patients [68]. Wang et al. also reported reduced levels of spermidine to be implicated in glaucomatous damage [69].

### 3.4. Fisetin

Fisetin is a flavonol, which is part of a group of compounds known as polyphenols. It is derived from common fruits and vegetables such as apples, onions, and cucumbers. It is also found in eudicotyledon trees and shrubs. It has a distinctive yellowing effect and is therefore used as a dye. The biochemical precursor is phenylalanine. Like resveratrol, studies have shown that fisetin can prolong the life of lower animals [70].

Fisetin has multiple beneficial pathways that can help in disease reduction. These include neuroprotective and anti-angiogenic properties. The anti-proliferative abilities of this substance work at interfering with the cell cycle through multiple channels. It inhibits the PI3K/Akt and mTOR pathways in humans and is therefore important for managing prostate cancer [71]. The anti-carcinogenic activity of fisetin may be because it is a topoisomerase inhibitor [72]. It also mediates its anti-cancer properties by senolysis, which is said to be twice as potent as quercetin [73]. Fisetin inhibits colon cancer cells by suppressing COx2 and Wnt/EGFR/NF-kappaB signaling pathways. Fisetin also suppresses the p38 MAPK-dependent NF-kB signal pathways, leading to a downregulation of the urokinase plasminogen activator [74].

Fisetin essentially has a wide range of anti-cancer capabilities [75] in addition to other chronic diseases [76]; it also has anti-inflammatory and neurotrophic abilities [77]. Kubina et al. reported cytotoxic activity of fisetin against cancer cells [78]. Lall et al. also attributed its anti-carcinogenic properties to apoptotic cell cycle dysregulation [79]. Fisetin has been shown to prevent the growth and spread of cancers of the breast, cervix, and colon [80,81,82,83]. Prostrate-specific antigen levels in athymic nude mice were found to be reduced after treatment with fisetin supplements [84].

Fisetin has shown extensive uses in ophthalmology. Vascular endothelial growth factor (VEGF) activation in diabetic retinopathy (DR) results in the proliferation of rogue blood vessels. Fisetin downregulates VEGF production, thereby suppressing the progression of DR [85]. The anti-inflammatory actions of this substance have been shown to protect retinal functions in glaucomatous murine models [86]. Fisetin has also shown anti-carcinogenic properties by stimulating mitochondrial apoptosis in uveal melanoma cells [87]. In an experimental study by Kan et al., a fisetin-treated group was found to have statistically lower levels of cataracts compared with alpha-lipoic and placebo-treated mice groups in a study where diabetic cataracts was being induced [88].

### 3.5. Omega-3 

Omega-3 polyunsaturated fatty acids are termed with this name based on structural features. Fatty acids possess a carboxylic chain on one end, which is designated as ‘alpha’; meanwhile, a methyl group exists at the ‘omega end’. For omega-3s, the cis double bonds are first separated by a methylene group on the third carbon atom of the omega end.

Omega-3s are a group of polyenoic fatty acids (FAs) that includes both long-chain and short chain substrates, such as: DHA {cis-4,cis-7,cis-10,cis-13,cis-16,cis-19- docosahexaenoic acid};DPA {cis-7,cis-10,cis-13,cis-16,cis-19- docosapentaenoic acid};EPA {cis-5,cis-8,cis-11,cis-14,cis-17- eicosapentaenoic acid}; as well asSDA: stearidonic acid {cis-6,cis-9,cis-12,cis-15- octadecatetraenoic acid} and;Short-chain ALA: alpha-linolenic acid {cis-9,cis-12,cis-15-octadecatrienoic acid}.

They are mainly found in oils derived from the liver and integument of aquatic mammals. However, plant-like organisms such as microalgae and other forbs have been theorized to synthesize compounds with similar biochemical structures. Dietary sources from oily fish (cod, anchovies, salmon, herring, mackerel, etc.) can provide modest levels of EPA and DHA [89]. ALA is obtained from flaxseed and nuts [90]. Numerous biological properties of polyunsaturated fatty acids are yielded via lipid mediators of fatty acid oxygenases, which include cytochrome P450 mono-oxygenases, cyclo-oxygenase, and lipo-oxygenase enzymes [91].

Long-chain polyunsaturated FAs, especially DHA and EPA, possess anti-inflammatory effects. Omega-3 FAs lend an anti-inflammatory effect via inhibition of the phospholipase A2 released from stressed or injured/inflamed tissue. In addition, these substances compete with eicosanoid formation that is caused by omega-6 arachidonic. This subsequently results in reduced production of the eicosanoids (i.e., prostacyclin, cytokines, leukotrienes, and prostaglandin-E1) [91]. Vaso-protective and cardioprotective effects of dietary omega-3 FA intake may be attributed to the cytochrome P-dependent mechanisms of EPA and DHA [92].

The consumption of omega-3s has shown potential effects in the prevention of neurodegenerative, cardiovascular, and neoplastic diseases. EPA and DHA are beneficial for managing inflammatory and autoimmune conditions such as rheumatoid arthritis [93] and osteoarthritis by modulating the immune response. Thus, omega-3 PUFA supplementation is recommended for hypertriglyceridemia [94]. This nutrient also helps to maintain the integrity of the vascular endothelium. It propagates normal growth of the vascular wall in the fetal system. Studies have demonstrated that several neural support structures, including cerebral grey matter, the CNS [95], and tissue within the retina, are composed of phospholipids that are derived from ALA.

DHA and EPA have been linked with decreased severity and progression of diabetic retinopathy (DR) among individuals with diabetes mellitus type 2 (NIDDM) [96] due to their activities against inflammation. Omega-3 polyunsaturated fatty acids may also someday play a more central role in managing corneal neuropathy that is secondary to local or systemic disease [97,98]. Systemic (oral) omega-3 fatty acid supplementation has also proved its therapeutic significance in the management of dry eye disease (DED), particularly in DED that is secondary to meibomian gland dysfunction [99]. Omega-3 FA supplementation is equally efficacious for managing dry eye symptoms triggered by the chronic instillation of antiglaucoma drops [100]. Subsequently, management possibilities using the innovative topical administration of the omega-3 FAs have been assessed [101]. It has also been suggested that an adjunctive antioxidant and multivitamin supplementation with docosahexaenoic and eicosapentaenoic fatty acids decreases the risk of progression from early to late age-related macular degeneration (ARMD) [102,103].

Topical docosahexaenoic acid showed inhibitory effects on bleb fibrosis to a degree that was comparable to mitomycin C in murine eyes following glaucoma filtration surgery [104]. Cellular oxidative stress is considered to be an underlying factor influencing the degeneration of trabecular meshwork endothelial cell degeneration and subsequent chronic IOP elevation [105,106,107]. Polyunsaturated fatty acids are strongly suggested to offer retinal ganglion cell neuroprotection and antioxidant effects in glaucoma [108,109]. The retinal neuroprotective effects of DHA and EPA in combination with timolol have been reported in studies examining treatment strategies for hereditary glaucoma in murine models [110]. Similar neuroprotective effects via omega-3s have also been observed for murine hereditary optic atrophy and anterior ischemic optic neuropathy, respectively [111,112]. There may also be potential benefits to intraocular pressure (IOP) management in ocular hypertension (OHT) and normal tension glaucoma (NTG) [30].

### 3.6. Rapamycin 

Rapamycin is a macrolide (polyketide) compound derived from Streptomyces hygroscopicus. Rapamycin (RAPA) is a specific inhibitor of the mTORC1 signaling pathway. This substance modulates neuroglial proliferation via the inhibition of ‘pro-proliferative’ mTORC-1 signaling [113]. The depicted benefits applicable in neoplastic and degenerative diseases are provided by the antioxidant properties via the enzymatic inhibition of nitric oxide synthase and de-catalyzation of the release of ‘free’ reactive oxygen species (ROS). The macrolide chemical structure of rapamycin provides antimicrobial (antibacterial and antifungal), anti-inflammatory, and immunomodulatory properties [114]. By hindering the activation of activator protein 1 and nuclear factor kappa B (NF-kB), which mediate the induction of cyclo-oxygenase-2 (COX-2) enzyme and other inflammatory cytokines, mTOR inhibition consequently reduces inflammatory responses at the cellular level.

RAPA promotes immunomodulation via the inhibition of macrophage cell function and cellular migration following exposure; it also possesses several immunosuppressive properties. RAPA acts on interleukin-2 (IL-2) and costimulatory signaling pathways, thus attenuating the immune response. mTOR governs the cellular pathways of apoptosis, autophagy, and necroptosis [113]. Inhibition of mTOR by rapamycin suppresses the immune response by preventing cell cycle progression from the G1 phase to S phase, thereby blocking proliferation [115].

RAPA has gained clinical significance in the field of anti-aging medicine based on animal-based research outcomes, which have suggested that a dose-specific effect on lifespan extension can be elicited following acute administration of the mTOR inhibitor during early stages of life [114]. RAPA has also been reported to yield benefits in the setting of neurodegenerative disease management. It has been suggested that this effect occurs as a product of cellular autophagy modulation [116]. The potential value for understanding Alzheimer’s disease has been linked to the possible mechanisms of autonomic neuron mitochondrial dysfunction, known as mitophagy [116], and other related neuroinflammatory processes. mTOR inhibitions via RAPA and related analogs have been linked to increased survival rates among patients with metastatic renal malignancies [117]. The pathway of mTOR inhibition may also provide novel strategies for managing autosomal dominant polycystic kidneys [118]. Evidence-based data also recommends mTOR inhibition for managing tuberous sclerosis complex-associated epilepsy [119]. The mechanistic role of RAPA has shown potentially promising results for managing a wide range of hepatic pathologies [120]. RAPA is also widely used to prevent organ transplant rejection.

The immunosuppressive mechanisms of RAPA have been reported to ease manifestations of refractory ocular inflammatory syndrome, including autoimmune dacryoadenitis and Grave’s orbitopathy [121,122,123]. The immunomodulatory effects are also applied for the prophylaxis of corneal allograft rejection. mTOR has been linked with corneal neovascularization in vitro [124]. Anti-proliferative properties of RAPA have been observed to prevent the proliferation of human retinoblastoma cells [125]. General strategies targeting the mTOR signaling pathway have been determined to possess viable therapeutic potentials for managing ocular neurodegenerative disorders [126]. The in vivo application of RAPA was correlated with the partial reversal of metabolic failure via the loss of complement factor H in retinal pigment epithelial cells, which is an important pathophysiological contributor to photoreceptor degeneration in ARMD [127]. mTOR inhibition via systemic administration of RAPA also has been shown to suppress VEGF expression and retinal oxidative stress response to hyperglycemia in murine models [128]. RAPA has also been shown to counteract senescence and apoptosis of human corneal epithelial cells [129].

Regarding glaucoma, topical application of RAPA has been associated with decreased intraocular pressure via the added inhibition of Rho-associated protein kinase in a murine aqueous humor [127]. mTOR inhibition by RAPA has also been found to provide neuroprotection for retinal ganglion cells (RGCs) [130]. These neuroprotective effects are also inherent following ischemic retinal injury [131]. mTOR-modulated retinal pericyte stabilization has also been observed in vitro [132].

### 3.7. Metformin 

Metformin is a biguanide compound originally derived from Galega officinalis (G. Officinalis), the French lilac [133]. Hepatic transport of metformin occurs via the hepatic portal vein following intestinal absorption.

OCT1 is a hepatic uptake transporter. OCT1 and PMAT are also reported to play combined roles in the intestinal accumulation of metformin. The exact biochemistry of intestinal OCT1 remains unclear; however, PMAT has been known to be expressed in the apical membrane of enterocytes. Metformin is hypothesized to primarily act on hepatic metabolism. Metformin acts via adenosine monophosphate-activated protein kinase (AMPK)-dependent and AMPK-independent mechanisms. The therapeutic effects have been linked to OCT1 activation of AMPK.

Metformin exerts its effects of glycemic control by acting on the liver via AMPK activation and by inhibiting hepatic glucose production (HGP)/gluconeogenesis [134]; it also improves muscular and hepatic absorption of glucose substrates. Due to its favorable safety profile, metformin is considered an essential oral antidiabetic for managing diabetes mellitus type II (NIDDM). For non-diabetic subjects, it has been found to attain significant plasma concentrations within three hours of administration. Serum concentrations reach nearly two-fold values in diabetic individuals who are on consistent daily doses of about one gram of metformin. The weight loss-inducing effect [134] is also suggested to modulate cardiovascular risk factors among individuals with NIDDM.

Mitochondria-targeted therapy via metformin has been considered a potential strategy for reducing oxygen demand and consumption in neoplastic lesions [135]. The anti-proliferative attributes of metformin have been established via the inhibition of breast cancer cells in vitro as well [136]. Metformin has also found versatile applications in obstetrics and gynecology. This substance has gained attention for managing gestational diabetes and polycystic ovarian syndrome [137].

In the field of ophthalmology, metformin has been found to yield control of steroid-induced ocular hypertension in mouse models [32]. This outcome has been suggested to be associated with a metformin-dependent exertion of antioxidant effects on the trabecular meshwork cytoskeleton via the integrin/ROCK pathway [32]. The antioxidant mechanism of metformin has also been reported to include the diminution of reactive oxygen intermediates within the rod photoreceptor outer segments in response to light absorption [138]. Inhibition of hypoxia-inducible factor-1alpha via metformin has been hypothesized as a plausible novel therapy for retinal degenerative disease [139]. A possible association between metformin treatment and a lower risk of open-angle glaucoma among individuals with NIDDM has been suggested [140,141]. Over the last decade, potential therapeutic benefits of metformin and several other biologic agents have been hypothesized in the context of non-exudative ARMD [142]. Metformin therapy in NIDDM has yielded no correlation with an increased risk or accelerated onset of age-related macular degeneration [143].

### 3.8. Alpha-Ketoglutarate 

Alpha-ketoglutarate (aKG) is a precursor of glutamate and glutamine. Glutamate is yielded by adding an amino group to the aKG molecule. The conversion of aKG to glutamine within the plasma membrane is catalyzed by glutamate dehydrogenase (GDH) and glutamine synthetase (GS) via the addition of a nitrogen group. aKG is an intermediate metabolite substrate of the tricarboxylic acid cycle (TAC, which is essential for cellular metabolism [144].

aKG possesses a strong affinity for reactive oxygen species (ROS) released from the mitochondria, especially hydrogen peroxide, which it reacts with as a result of the conversion of succinate, carbon dioxide, and water, subsequently leading to its elimination [145]. Excess-free/unbounded ROS such as superoxide anions, hydrogen peroxide, etc., inhibit normal deoxyribonucleic acid (DNA) synthesis. They consequently cause damage to the cytoskeleton and tissue integrity. aKG is thus a potent antioxidant that enables tissue repair.

aKG administration has been correlated with increased bone among aged experimental animals [143]. The use of this substance has also been associated with improved potency in the bone marrow-derived stem cells (MSCs) obtained from senescent mice [146]. aKG supplementation has been linked with improved cardiac function in mice with pre-existing myocardial dysfunction [147]. aKG supplementation has been reported to suppress the growth of colorectal cancer tumor models via inhibited Wnt signaling mechanisms [148]. aKG-dependent circular RNA (circFTO) has been reported to have pro-angiogenetic effects and impair the blood–retinal barrier in DR. aKG-modulation strategies targeting at-risk tissues may thus potentially help regulate DR progression [149,150]. The aKG dehydrogenase complexes play a key role in intracellular glucose metabolism [151]. Studies have also suggested that the antioxidant effects of aKG include reduced cataractogenesis in experimental animal models [152].

### 3.9. Vitamin B3 (Niacin)

Niacin (also known as nicotinic acid) is an essential, water-soluble form of vitamin B3 with contrasting effects when used at various dosages. This organic compound is a pyridine derivative with a carboxyl group at the 3-position. Niacin is a precursor to NAD+/NADH and NADP+/NADPH, both of which are required for metabolic functions in living cells. It is involved in both DNA repair and adrenal gland steroid hormone production. Niacin has been confirmed to cause mild-to-moderate serum aminotransferase elevations.

Niacin is a cofactor in over 400 biochemical reactions in the body, including in metabolic processes. Julius [153] concluded that niacin plays a role as a replacement for statin and/or as an important additive in statin-intolerant patients. Patients with elevated triglyceride and low HDL cholesterol levels, in addition to patients with elevated lipoprotein (a) concentrations, can possibly reap benefits from using niacin. Xu and Jiang [154] published a study about the psychiatric manifestations of niacin deficiency and found that schizophrenia patients can benefit from niacin augmentation.

In the field of ophthalmology, Gaynon et al. [33] suggest that there are marked functional and anatomical improvements in patients with central retinal vein occlusion (CRVO) who are undergoing niacin therapy. This study also predicted the possibility of increased niacin inclusion for the management of CRVO. Charng et al. [155] reported that while no significant correlation can be found between dietary niacin intake and the age-related thinning of the retinal nerve fiber layer in healthy eyes, niacin in supraphysiological doses is found to have a therapeutic effect on the retina. However, despite the indication of niacin for treating hyperlipidemia, Domanico et al. [156] summarized that small doses of niacin precipitate the development of cystoid macula edema symptoms such as blurred vision, reduced visual acuity, and metamorphopsia experienced by patients.

### 3.10. Vitamin D

Vitamin D (ergocalciferol) is a fat-soluble vitamin that is a member of the calciferols, and it essentially functions as a hormone. Vitamin D was discovered to be crucial to the prevention of rickets. It is denoted as the most important vitamin and is derived as a benefit from normal levels of exposure to sunlight. Vitamin D is produced in the epidermis of animals due to the precursor molecule 7-dehydrocholesterol absorbing light quanta. Vitamin D needs to be converted to 1,25-dihydroxycholecalciferol, which is its active form. This transformation takes place in two steps: Cholecalciferol is hydroxylated to 25-hydroxycholecalciferol in the liver by the enzyme 25-hydroxylase. Then, within the kidney, 25-hydroxycholecalciferol acts as a substrate for 1-alpha-hydroxylase, resulting in the biologically active form 1,25-dihydroxycholecalciferol.

Vitamin D is a long-known vitamin that aids in the body’s absorption and storage of calcium and phosphorus. These two nutrients are especially important for bone formation. In oncology, researchers [157] have identified that vitamin D metabolism is important in the management of cancer patients. A growing body of evidence points to the dysregulation of vitamin D metabolism and functions in many cancer types, which confers resistance to the antitumorigenic effects of vitamin D and aids in the growth and development of cancer. It has also been reported in several studies that vitamin D can serve as an adjunct therapy in SARS-CoV2 virus infections [158]. Studies have shown that this vitamin can increase cellular immunity and induce antimicrobial peptides, hence reducing the severity of viral infection [159]. The mechanisms involved include elevating the levels of anti-inflammatory cytokines while lowering the levels of proinflammatory cytokines. It is also indicated as a therapeutic option in the management of diabetes mellitus [160,161] and skin diseases [162]. Cheng et al. [163] conducted a systematic review and randomized control trial and observed that vitamin D can reduce negative emotions. Supplementation with vitamin D helps lessen unpleasant feelings. It is posited that beneficiaries of supplementation are people with major depressive illness and those who are vitamin D deficient. Chan et al. [164] showed that this vitamin is an important molecule that can have negative effects on individuals showing deficiency.

Concerning ophthalmology, studies have shown strong relationships between vitamin D deficiency and myopia development, glaucoma, age-related macular degeneration, diabetic retinopathy, and dry eye syndrome [25]. Studies by Arikan and Kamis [165] have shown that vitamin D has regulatory effects on non-skeletal issues, including neurons, which can affect the contrast sensitivity function and general quality of vision. Akkaya and Ulusoy [166] researched keratoconus and vitamin D levels. They showed that keratoconus patients had lower vitamin D serum levels than their age and sex-matched control groups. This may explain vitamin D’s prognosis to keratoconus development and outcomes.

### 3.11. Zeaxanthin

Zeaxanthin is a carotenoid molecule with a conjugated double-bond system, which gives it its characteristic yellow color. It consists of a central chain of 40 carbon atoms, with two hydroxyl groups (-OH) on either end. Zeaxanthin melts at about 215–216 °C. It is insoluble in water, but it is soluble in polar organic solvents such as ethanol, acetone, and chloroform. Zeaxanthin is synthesized through the methylerythritol phosphate (MEP) pathway, also known as the non-mevalonate pathway, which is an alternative biosynthetic pathway for isoprenoid compounds in bacteria, plants, and algae. The MEP pathway involves seven enzymatic steps that convert pyruvate and glyceraldehyde-3-phosphate to isopentenyl pyrophosphate (IPP) and dimethylallyl pyrophosphate (DMAPP), the precursors for carotenoid synthesis. It can be converted to other carotenoids through a variety of metabolic pathways. For example, zeaxanthin can be converted to violaxanthin, antheraxanthin, and neoxanthin through a series of enzyme-catalyzed reactions that are known as the xanthophyll cycle. This cycle plays a crucial role in regulating the amount of light absorbed by chloroplasts in plants, which helps protect them from excess light and prevents photodamage. Zeaxanthin has strong absorption in the blue-green region of the visible spectrum and reflects yellow light. It exists in two isomeric forms: (3R,3’R)-zeaxanthin and (3R,3’S)-zeaxanthin, which differs only in the orientation of the hydroxyl groups on the end rings.

Zeaxanthin possesses anti-inflammatory effects that can reduce the risk of chronic diseases such as heart disease, cancer, and arthritis. It enhances the activity of white blood cells, which play a crucial role in fighting infections and diseases. Furthermore, zeaxanthin improves cognitive and cardiovascular health by reducing oxidative stress and inflammation. Some studies have suggested that zeaxanthin has benefits for skin health. A double-blind, placebo-controlled study found that supplementation with a combination of zeaxanthin and other carotenoids improved skin hydration and elasticity in healthy middle-aged women [167]. Another study found that zeaxanthin reduced the number of sunburned cells in the skin after UV irradiation in mice [168]. A study in healthy elderly men found that supplementation with lutein and zeaxanthin improved immune function by increasing the activity of natural killer cells [169]. Moreso found that zeaxanthin reduced cognitive decline in mice with Alzheimer’s disease [170]. Kishimoto et al. conducted a clinical trial and reported that supplementation with lutein and zeaxanthin reduced oxidized LDL cholesterol levels in people with metabolic syndrome [171].

The benefit of lutein to the eye is to act as a filter that protects the retina from blue light and oxidative damage [172]. It also has antioxidant properties that help protect the skin from UV radiation damage and oxidative stress [173]. It may also improve skin elasticity and hydration. A randomized controlled trial found that supplementation with lutein and zeaxanthin reduced the risk of progression to advanced age-related macular degeneration by 25% for people with early-stage macular degeneration [174]. Another study found that supplementation with lutein and zeaxanthin improved visual function in people with early age-related macular degeneration [175]. Loughman et al. evaluated fovea pigment response in glaucoma-affected individuals to supplementation with zeaxanthin among other nutrients and concluded that there was a significant improvement after supplementation. their results were in line with data from Lew et al. [176] and Sanz-González et al. [174]. Zeaxanthin may also slow down the rate of cataractogenesis [177,178].

### 3.12. Lutein

Lutein is a carotenoid pigment that is naturally found in various fruits and vegetables, particularly in leafy green vegetables such as spinach and kale [179]. It is a yellow-to-orange colored pigment that is insoluble in water but soluble in organic solvents such as ethanol, hexane, and chloroform. Lutein is known for its antioxidant [180,181] and anti-inflammatory properties [182], and it has been associated with numerous health benefits, particularly for eye health. It is one of the major carotenoids found in the human retina and is believed to protect the eye from age-related macular degeneration and other eye diseases. Overall, lutein has several biological properties that make it a beneficial compound for human health, particularly for eye health, skin health, and cognitive function [183,184].

Lutein is a powerful antioxidant that can neutralize free radicals and reactive oxygen species, which can cause oxidative damage to cells and tissues. It can help protect cells from oxidative stress and reduce the risk of chronic diseases such as cancer, cardiovascular disease, and neurodegenerative disorders [185]. Chronic inflammation is associated with numerous health problems, including arthritis, asthma, and heart disease. The anti-inflammatory properties of this substance can help prevent and manage these conditions [186,187]. Lutein has been shown to regulate the expression of genes involved in inflammation, oxidative stress, and other biological processes. It has been reported to upregulate the expression of antioxidant enzymes such as superoxide dismutase and catalase. Lutein modulates signal transduction pathways that are involved in inflammation, oxidative stress, and cell proliferation [188]. This compound inhibits the activation of nuclear factor kappa B (NF-kB), a transcription factor involved in the expression of pro-inflammatory cytokines [189,190]. Lutein is implicated in the modulation of lipid metabolism, which may contribute to its beneficial effects on skin health. It can increase the expression of genes involved in lipid synthesis and may also inhibit the expression of genes involved in lipid oxidation [191].

Concerning ophthalmology, lutein accumulates in the macula, acting as a filter for harmful blue light and helping to protect the eye from oxidative damage [192]. Lutein also acts via antioxidative and barrier properties to protect the retina photoreceptors, as seen in the work of Obana et al. [193]. It has been shown to reduce the risk of age-related macular degeneration, glaucoma, cataracts, and other eye diseases [178,194,195]. A systematic review concluded that lutein-supplemented eyes performed better due to their neuroprotective properties and preservation of synaptic activity [176]. Fung et al. induced apoptotic and autophagic stress in rat muller cells using cobalt II chloride as a noxious agent to create a model for the pathophysiology of diabetic retinopathy. They subsequently showed that lutein attenuated the apoptosis and autophagic response [196].

### 3.13. Resveratrol

Resveratrol is a polyphenol that is naturally known to have a trans-stilbene structure, which was described when it was found in its natural state in red wine. This compound has two phenolic rings that are bonded together. The structure of resveratrol is made by the combination of two phenolic rings that are doubly bonded by a styrene bond that is doubled; this forms 3,5,4′-trihydroxystilbene with a molecular weight of 228.25 g/mol. This double bond is responsible for the isometric cis and trans forms of resveratrol. It is key to know that the trans isomer is the most stable form, and this is responsible for its medicinal or therapeutic effects because this isomer enhances its ability to bind to many biological molecules [197].

The benefits of resveratrol were first discovered after the “French Paradox”. Studies regarding this situation reported that red wine was a cardio-protective supplement for individuals in the Northern part of France, in which diets were high in saturated fat but mortality was low from heart diseases in comparison to other countries with a similarly high intake of fat in the diets. Studies have shown that phenolic compounds found in grapes and red wines exhibit a protective effect against diseases [198,199].

Resveratrol has unique chemical and physical features, which enhance its ability to cross cell membranes passively as well as interact with the membranes of cell receptors. This is responsible for its interaction with intracellular and extracellular molecules. The mode of action at the cellular level is initiated by signals from pathways in relation to the membranes of cells, triggering intracellular mechanisms or expressing itself in the core of the nucleus. Diseases such as cardiovascular problems, neurodegenerative diseases, cancers, and diabetes involve oxidative damage in their pathogenesis. Resveratrol is known to have antioxidant activity by disrupting the state of respiratory processes in the mitochondria, as found in a study conducted on rat brain mitochondria. It has also been found to initiate competitive substrate inhibition on coenzyme Q by reducing the concentration of complex III [200]. Resveratrol is also known to have phytoestrogen properties in that it binds the estrogen alpha receptors and beta receptors (ER-alpha and ER-beta). Studies have shown that ER-α is stereo-selective and has more affinity for the trans-isomer of resveratrol.

Resveratrol is a potent antioxidant. Studies have shown that its derivatives have a powerful inhibiting effect on inhibiting low-density protein induced by copper ions [201]. Resveratrol has also shown anticancer effects in in vitro and in vivo studies, where the evidence indicates that resveratrol inhibits initiation, promotion, and progression (carcinogenesis stages) [202,203,204]. Evidence from several other studies has shown that resveratrol displays therapeutic properties that are of medical importance, which include anti-inflammatory and anti-proliferative effects [205,206].

Resveratrol has a wide range of protective effects on the eye. It has an anti-cell death benefit (anti-apoptosis), antitumorigenic, anti-angiogenic, and other vascular protective properties [207]. Van et al. have shown the neuroprotective effect of resveratrol on ischemic damage to the retina in an experimental study on adult rats, in which comparisons were made between the control and treatment groups. The animals treated with resveratrol showed less ischemic injury [208]. A study conducted by Luna et al. on the effect of a dietary supplement on markers of glaucoma in the trabecular meshwork (TM) showed decreased longstanding oxidative stress, which was identified by the reduction of the RO produced as well as decreased levels of inflammation indicators such as interleukins and endothelial-leukocyte adhesion molecule-1 [209]. In another study conducted by Prihan et al. [210], the authors explored the role of giving resveratrol alone and a combination of riluzole and resveratrol in a rat model with glaucoma by assessing the retinal ganglion cells (RGCs). Glaucoma was induced in the early and late stages. The results suggested specific effects of resveratrol, showing neuroprotection effects on the RGCs. Resveratrol is known to prevent induced disorders from oxidative stress and sodium iodate-induced apoptosis in retinal pigment epithelium (RPE) cells in an in vitro setup [211]. Resveratrol could be beneficial when managing cases of macular degeneration, considering that it has neuroprotective properties. Several studies have also shown the therapeutic effect of resveratrol on other eye-related diseases such as DR [212]. The protective role of resveratrol has been reported in several different ocular tissues [213,214,215].

### 3.14. Pyruvate

Pyruvate in the mitochondria is responsible for the mitochondrial pyruvate carriers, which mediate pyruvate import into the mitochondria. This is important in the major oxidative processes in the mitochondria such as the tricarboxylic cycle and oxidative phosphorylation. Inhibiting this process of pyruvate carrier-mediated pyruvate transport has been found to be beneficial and protective in neurodegenerative diseases such as Parkinson’s disease [216]. Pyruvate biochemical constituent is a 2-oxo monocarboxylic acid anion that is the conjugate base of pyruvic acid, which is an important metabolic product for energy-producing biochemical pathways. Pyruvate is the end product of glycolysis, and in a state of hypoxia, it can be metabolized to form lactate anaerobically. Pyruvate is a precursor of acetyl-coenzyme A (AcetylCoA) and oxaloacetate, which are involved in the tricarboxylic acid (TCA) cycle [217].

Pyruvate is one of the end products of glycolysis, and the transport mechanism that enhances pyruvate transport into the mitochondria is known as the mitochondrial pyruvate carrier. This is important in major oxidative processes in the mitochondria, such as the tricarboxylic acid cycle and in commonly biochemical reactions such as oxidative phosphorylation. This biochemical compound has been known to be protective against neurodegenerative diseases such as Parkinson’s disease [218]. Pyruvate is a major molecule that is important for several aspects of cellular and human metabolism. Pyruvate formed from glycolysis is the outcome of cellular cytoplasmic addition that is ferried to the mitochondria as one of the major sources of energy in the tricarboxylic acid cycle. Pyruvate is a cornerstone for ATP generation in the mitochondria and for making many of the major pathways intersecting the citric acid cycle effective [219]. The availability of pyruvate in the cytosol depends on the availability of enzymes such as pyruvate kinase, lactate dehydrogenase, and alanine aminotransferase. Cytosolic pyruvate needs to be actively moved into the mitochondrial matrix once formed, and the means of intermembrane transportation is through the mitochondrial pyruvate carrier (MPC) [219]. It is a known biochemical fact that the mitochondrial inner membrane is impermeable to ionic molecules even though other biochemical molecules and pyruvate can freely diffuse from the cytoplasm through porins [220,221]. The MPC found in humans is formed by two very similar subunits, which are known as MPC1 and MPC2. Their stoichiometry and physiological control and regulations are not well understood.

Endogenous pyruvate is key in aerobic and anaerobic respiration and is a significant molecule in the production of energy; exogenous pyruvate is shuttled into the cells through monocarboxylate transporters and mostly functions as an antioxidant. Exogenous pyruvate is reduced in conditions such as hyperglycemia, nephropathy, and retinopathy in animals with induced diabetes. In vitro studies on cultured rat cortical neurons through a mechanism related to monocarboxylate transport carriers against apoptosis have shown the ability of exogenous pyruvate to inhibit neuronal cell death [222]. Pyruvate has been reported to have an edge over lactate in peritoneal dialysis [223]. Another study also showed that it potentiates neurotrophic oxide generation in peritoneal dialysis solutions [224]. Pyruvate has rare therapeutic characteristics for modifying and correcting abnormalities of the neuronal network due to phagocytic actions against neuronal network abnormalities during nerve tissue disease. It is known that reductions in the concentration of NAD+ in cells can lead to excessive activation of poly adenosine diphosphate-ribose (PARP-1), which leads to cell death due to the absence of energy. Pyruvate hinders the overactivation of PARP-1 [225]. In addition, pyruvate is a direct substrate for mitochondrial metabolism, and its oxidation does not depend on the cytoplasmic redox state. Pyruvate bypasses the restrictions imposed by PARP-1 and can restore energy deficiency in such conditions. It reduces the blood glutamate level, which is targeted in the treatment of most neurodegenerative diseases, enhancing the glutamate passing out of the brain tissue through the ion-selective blood–brain barrier, thereby reducing glutamate-induced neurotoxicity [225]. Pyruvate is known to increase neuronal tolerance as it augments glycogen stores, which helps in neuronal ischemic and hypoglycemic conditions. Apart from the known functions in energy metabolism, it has therapeutic neuroprotective functions, which are widely untapped in treatments.

A study has been performed using RNA-sequencing through a large-scale study (metabolomics) to examine early glaucoma in mice. The findings in the study have shown that there was neuroprotection from cell death in rat and mouse models of glaucoma that were fed with oral supplements of pyruvate [130]. Other clinical trials used nicotinamide and pyruvate in a phase II clinical trial and yielded significant momentary improvement in visual function, supporting the findings in the former experimental research and indicating a role for these agents in neuroprotection for individuals with glaucoma; this emphasizes the necessity for lasting studies to establish the longitudinal therapeutic effects of these nutrients [226], even if glaucoma has traditionally been exclusively all about pressure [227].

### 3.15. Vitamin A 

Vitamin A (retinol) is a fat-soluble nutrient. It has been proven to have a neuroprotective capacity. This function is associated with the all-trans form of retinol, which has been shown in experimental studies conducted on rats [228]. It is known to have diverse characteristic neuroprotective potentials, although high doses of vitamin A (retinol) can be harmful and responsible for raised intracranial pressure, anorexia, and congenital malformations in early pregnancy [229]. According to the International Union of Biochemistry and Molecular Biology (IUBMB), retinol and other compounds such as retinoic acid and retinoic aldehyde are classified as retinoids. Retinoids are all derivatives of vitamin A, which could be natural or synthetic forms of vitamin A that do not have an aromatic portion of β-ionone [230].

Several physiological processes have shown the vital roles of vitamin A, including growth, tissue development at the embryogenic stage, proper development and functioning of the reproductive organs, proper functioning of the immune system, and the commonly known function of enhancing vision in the eyes. As a lipid-soluble vitamin, retinol stimulates retinoid receptors (RARs), the stimulation of which initiates the differentiation of cells as well as induction cell death of tumorigenic or cancerous cells, hence hindering the development of cancers. Because of the insolubility in body fluids, retinol is transported by specialized proteins known as retinol-binding proteins (RBP) through a complex involvement with transthyretin [231]. Cytosolic retinol-binding protein (CRBP) is in the cytoplasm and is known to have a high affinity for retinol. CRPB has two subtypes of receptors known as CRPB I and CRPBII [232]. Several studies have shown that the antioxidant properties of retinol exist through the mechanism of blocking the voltage-gated calcium channels, and this single mechanism is neuroprotective despite the fact that retinol has many biological effects [233,234].

A study conducted by Sato et al. examined the neuroprotective ability of vitamin A (particularly the trans retinol) in a mice model and found that there was optimum protection in the treatment group compared with the controls [235]. This study showed that retinol had neuroprotective potential. Other studies have reported the ability of vitamin A to act as an anti-ageing agent in skin cosmetics [236]. The viability of vitamin A in preventing neuroinflammation and neurodegeneration in Alzheimer’s disease has also been documented in the current literature [237].

Vitamin A is an ancient treatment for nutritional blindness, including for cases of xerophthalmia, keratomalacia, and the ocular manifestation of hypovitaminosis A. Dietary supplementation and fortification are recommended to retard the widespread prevalence of vitamin A deficiency [238]. Studies have recommended that vitamin A could be useful for managing retinal degenerative diseases such as age-related macular degeneration and Stagardt’s disease [239]. Vitamin A can also been found in several oral supplements together with other substances in the management of glaucoma.

### 3.16. Vitamin B1

Vitamin B1, also known as thiamine, is a water-soluble vitamin made up of two heterocyclic rings, which include a thaizole and pyrimidine ring interconnected with methylene. Thiamine is relatively stable in acidic solutions but reacts to heat, oxygen, and light. Biologically, thiamine is an important coenzyme for the completion of processes such as converting pyruvate to acetyl-CoA in the citric acid cycle to produce ATP, carbon dioxide, and other reducing agents. As a coenzyme in this process, thiamine is present in its active form, thiamine pyrophosphate (TPP).

The benefits of thiamine in medical disorders include its use in treating Wernicke–Korsakoff Syndrome. High doses of thiamine supplementation have been shown to improve cognitive function and reduce further neurological damage [240]. Heart failure has been linked to thiamine deficiency. A study by Rauchhaus et al. concluded that thiamine supplementation improved left ventricular ejection fraction and exercise capacity in heart failure patients with thiamine deficiency [241]. Day et al. reported that treating patients with alcohol withdrawal symptoms with thiamine produced positive results as it reduced the severity of symptoms [242].

Thiamine supplementation has been shown to improve visual function and reduce the risk of disease progression in patients with early-stage age-related macular degeneration, as supported by the observed reduction in oxidative stress markers in the retina [243,244]. When thiamine is supplied in the right doses, it has been shown to help reduce the risk of developing diabetic retinopathy [245]. Thiamine has also been shown to reduce the incidence, severity, and intensity of cataracts [246]. In a study conducted by Lee et al., a connection was found between thiamine intake and a reduction in disease progression for glaucoma patients with normal tension glaucoma [247].

### 3.17. Vitamin B2 (Riboflavin)

Riboflavin is a water-soluble vitamin that is relatively stable in heat and acid but sensitive to light. It serves as an important coenzyme in various metabolic processes such as in the conversion of food into energy and in the production of amino acids and fatty acids. Riboflavin shows an important role in cell repair and growth, considering that it helps eliminate free radicals that are capable of cell damage and that are capable of developing chronic diseases.

Riboflavin is a precursor for coenzymes such as flavin mononucleotide and flavin adenine dinucleotide, which serve as electron carriers that are important for metabolic pathways. It is also involved in the metabolism of iron to produce hemoglobin as it converts iron from its ferric state to ferrous, which is the active form in red blood cells (RBCs). Riboflavin is actively involved in the synthesis of opsin, which combines to form rhodopsin, a visual pigment required for low-light vision.

With respect to the use of riboflavin in clinical research, Wangsuwan et al. conducted research into the influence of riboflavin on acne vulgaris and concluded that vitamin B2 supplements significantly reduced the severity of acne lesions and improved the general skin health of patients in the study [248]. Baquerizo-Nole et al. concluded that riboflavin reduces rosacea symptoms [249]. Riboflavin has been studied for potential adjunct therapy in cancer management by Wang et al. [250]. The authors concluded that riboflavin can help inhibit the growth and spread of cancer cells and can serve as part of possible treatment regimens for cancer cases in the future. Bhide et al. also reported that riboflavin reduced the intensity of the side effects of cancer therapy (chemotherapy and radiotherapy) and generally improved the quality of life of cancer patients under care [251].

Kim et al. suggested that riboflavin may have a protective effect against the development and progression of age-related macular degeneration and glaucoma due to its antioxidative properties and active involvement in gene expression and cellular signaling pathways [252]. Riboflavin has also been studied by Raiskup et al., and they reported its role in strengthening corneal collagen fibers and improving the mechanical activities of the cornea in the treatment of corneal and other corneal abnormalities using corneal cross-linking therapy [253].

### 3.18. Vitamin B9

Vitamin B9, also known as folate, is a water-soluble vitamin and is one of eight essential B vitamins [254]. The chemical structure is similar to that of folic acid. Folate can be obtained from dietary sources, chiefly green leafy vegetables, as well as several animal sources (meat, eggs, etc.). The body easily absorbs synthesized sources of supplemental folic acid in greater amounts than it does from food-derived folate [255]. Within the body, oral vitamin B9 undergoes hepatic activation into tetrahydrofolate [256]. The primary route of excretion is via the kidneys.

Folate is key in the methylation processes, including in vitamin B12-associated methionine and homocysteine protein synthesis [257]. It also synergizes cobalamin to promote hematopoiesis (especially erythropoiesis) [258]. Via its role in DNA synthesis, vitamin B9 is essential for appropriate embryogenesis and fetal development [259].

Vitamin B9 has been attributed to have benefits in the setting of neurodegenerative diseases such as dementia due to its propagation of normal central nervous system development [260]. Folate supplementation has also shown positive effects in attenuation signs of toxicity and adverse effects secondary to oral doses of methotrexate, a disease-modifying anti-rheumatic drug [261]. It is routinely recommended during pregnancy for prophylaxis of embryonic or fetal neural tube defects [262]. Folic acid supplementation is occasionally used to reverse the metformin-induced reduction of serum folate and methionine concentrations [263]. Folate supplementation is occasionally employed alongside the long-term intake of sulfonamides [264]. These substances can be useful in the co-management of hemolytic anemia [265] and sickle cell anemia [266]. Folate supplementation has been associated with controlled risk of ischemic attacks among those with cardiovascular disease [267].

Vitamin B9 deficiency has been identified as a modifiable risk factor for hyper-homocysteinemic retinal diseases [268]. Murine-based retinal biopsies have also affirmed the aforementioned relationship [269]. Serum folate concentration has also been directly correlated with the risk of retinal vein occlusions [270,271]. Folate supplementation is grossly supported in acute cases of ischemic optic neuropathies or other hypoperfusion retinopathies, including in diabetes mellitus-related retinal ischemia [272,273,274].

Associations have also been found between elevated hyperhomocysteinemia, lower serum folate concentrations, and an incidence of pseudoexfoliation glaucoma [275]. However, no clear associations have been found between the incidence of either primary open-angle or normal tension glaucoma and serum folates or homocysteine levels [276]. One exception is in cases of primary open-angle glaucoma with endocrine systemic disorders and hematological anomalies (including folate deficiency) [277,278].

#### Vitamin B12 (Cobalamin)

Vitamin B12 is a water-soluble essential vitamin that is derived from dietary animal sources. Cobalamins possess a cyclic chemical structure comprising a cobalt element at its nucleus [279]. The two forms, which include adenosyl-cobalamin and methylcobalamin, serve as cofactors for the enzymatic action of methyl malonyl-mutase and methionine synthase, respectively [280]. Cobalamin absorption is enabled via intrinsic factors, transcobalamin, and haptocorrin proteins [279]. Hepatic storage accounts for reserves of vitamin B12.

Vitamin B12 and folates are regarded as cofactors for methylation reactions, particularly for DNA methylation and blood cell production [281]. Although transcobalamin and haptocorrin play extensive roles in serum vitamin B12 absorption, intrinsic factor production via parietal cells of the small intestine is essential for the intestinal absorption of dietary cobalamin [282]. Gastritis, gastrectomy, and ingestion of proton pump inhibitors reduce the gastric concentration of hydrochloric acid (HCl), which inhibits cobalamin release from food sources [283].

Individuals who have undergone bariatric surgery require intramuscular supplementation of vitamin B12 [284]. Long-term alcoholics also often develop deficiencies of intrinsic factors, leading to cobalamin malabsorption [285]. Chronic vitamin B12 malabsorption is a primary risk factor for developing pernicious anemia [286]. Vitamin B12 supplementation is routinely performed for individuals with type 2 diabetes as a long-term oral intake of metformin, which has been linked with the depletion of serum cobalamin levels [287]. Individuals on solely plant-based diets require B12 supplementation, as B12 is only obtained from animal-based foods [288]. Vitamin B12 deficiency has been associated with increased risk of CNS neurodegenerative changes, dementia, and peripheral neuropathy [289]. Its reported benefits in colorectal cancer patients are undetermined [290]. Combined folate and cobalamin supplementation reportedly improved hyperhomocysteinemia in murine studies [291].

Unlike vitamin B9, reduced serum vitamin B12 and elevated serum homocysteine values have been associated with PXG and other forms of open-angle glaucoma [292,293]. Impaired vitamin B12 absorption is associated with the onset and progression of nutritional optic neuropathy [294]. Chavala et al. also reported that B12 deficiency was correlated with the painless onset of central and cecocentral scotomas [295]. Cobalamin deficiency has also been correlated with the onset and resolution of optic neuritis, particularly among young individuals [296].

Vitamin B12 deficiency has been identified amongst etiologies of unilateral optic atrophy with contralateral disc edema without cranial space-occupying lesions (pseudo-Foster Kennedy syndrome) [297,298,299]. High rates of vitamin B12 deficiency have been reported among carriers of Leber’s hereditary optic neuropathy [300].

### 3.19. Vitamin C

Vitamin C is a water-soluble nutrient. It is a powerful component and catalyst for forming many neurotransmitters and it is a potent antioxidant [301]. While some animals can produce this vitamin autonomously, most mammals generally obtain their source from their diet due to the absence of the enzyme gluconolactone oxidase [302]. Vitamin C is easily destroyed by prolonged cooking. It is especially important for wound healing and collagen synthesis. A deficiency of vitamin C (hypovitaminosis) results in scurvy [303], a condition characterized by poor wound healing and a general breakdown of body tissues [304]. This condition may eventually become fatal if not properly managed [305]. A review by Maekawa et al. suggested that large concentrations of vitamin C in the blood, achieved through intravenous administration, exerted anti-cancer effects through oxidative mechanisms [306]. Gokce et al. have suggested that vitamin C deficiency is linked to body hair loss [307]. The exact mechanism is not yet understood. However, vitamin C is an established reproductive health supplement for men as clinical trials have shown improved sperm motility with vitamin C supplementation [308,309]. Marrof et al. showed that vitamin C may play a role in ameliorating symptoms of atopic dermatitis [310].

Cataracts largely results from the accumulation of free radicals [311]. Studies have shown that vitamin C supplementation can slow down cataractogenesis via its oxidative properties [312]. Gujral et al. showed that vitamin C was a potent supplement for enhancing wound healing [313]. As observed in other parts of the body, Vitamin C is equally important in regulating the integrity of ocular tissue, as it has been shown to regulate and even suppress VEGF formation [314]. Yuki et al. reported reduced levels of serum ascorbic acid in patients with normal tension glaucoma [315]. Similar findings of reduced plasma vitamin C were reported by Zanon-Moreno et al. [316]. These authors carried out a case-control study to assess generic risk score (GRS) using multiple genetic indices to compare 391 POAG patients with 383 healthy patients. They concluded that the GRS scores were inversely related to the plasma vitamin C levels, suggesting a reduced risk of POAG in patients with higher GRS scores and plasma vitamin C levels [317]. Han and Fu, however, reported a systematic review of databases that did not support the findings of Zanon-Moreno et al. and concluded that Vitamin C was not associated with a reduced prevalence of glaucoma [318].

### 3.20. Vitamin E

Vitamin E is a fat-soluble nutrient. These substances are from the classes of tocopherols (α, β, δ, γ isoforms) and trienols (α, β, δ, γ isoforms) [319,320]. It is a known antioxidant [321], immunomodulator [322], and antiplatelet compound [323]. The fat solubility of this vitamin means that it can easily be stored in body fat. Deficiencies in vitamin E are thus relatively rare [324]. Vitamin E has been widely reported to ameliorate immunosenescent changes [325,326]. Supplementation in the elderly with this substance has been shown to reduce the risk of respiratory complications in nursing homes [327]. Vitamin E also has effects on human reproductive health; an experimental study found that it increased sexual desire and satisfaction when administered together with ginseng in women [328]. It also has positive interactions in ameliorating certain skin disorders [329] and in cardiovascular medicine [330].

There is still much work to be conducted to understand the mechanism of action of vitamin E in relation to eye care. Clinical studies regarding the effects of this vitamin are still needed and are not conclusive [331,332]. These views have been confirmed in a large-scale systematic review by Matthew et al. [333]. Vitamin E has shown promise as an adjunct therapy in managing dry eyes [334]. Vitamin E-impregnated contact lenses have also been used as a platform for the delivery of the anti-inflammatory/anti-fibrotic drug Pirfenidone [335]. Another study reported that vitamin E-enriched contact lenses prolonged the action of timolol and dorzolamide that were infused into the same contacts [336]. Sekar and Chauhan confirmed similar results using vitamin E-impregnated contact lenses and prostaglandin analogs [337]. Vitamin E-enriched diets reduced RGC death in murine models compared with others on a normal diet [338]. Traces of vitamin E can also be found in supplements used in glaucoma management.

### 3.21. Citicoline

Citicoline is an isomeric form of choline, which is an intermediate metabolite of phospholipid synthesis within the cell membrane. Following administration, cytidine and choline are liberated from citicoline. It efficiently crosses the blood–brain barrier and reaches the CNS. Within the neuronal cell membrane, citicoline also serves as a choline donor in the pathway of neurotransmitter (acetylcholine) synthesis and activates biosynthesis of structural phospholipids, thus increasing brain metabolism [339,340].

Citicoline has been suggested to serve a neuroprotective function in ischemic diseases and reverse neural senescence in animal models. Citicoline has been reported to inhibit mechanisms of apoptosis that are associated with cerebral ischemia [341]. Its pharmacological characteristics and mechanisms suggest that the nutrient may be indicated for preserving motor function in the setting of cerebro-vascular disease, management of moderate-to-severe concussive head trauma, and cognitive disorders that follow post-concussion syndrome. Studies including patients who were managed with citicoline after suffering head trauma reported accelerated recovery from post-traumatic coma and other associated neurological deficits. Toxicological findings purport that citicoline demonstrates good safety profiles among human subjects [342,343].

Citicoline is currently being used as a supplement in glaucoma management. Citicoline has demonstrated activity in a range of central neurodegenerative diseases, and experimental evidence suggests its neuroprotective role traverses across retinal ganglion cells. Studies have reported associated improvements in visual function with the use of this substance [344,345]. Currently, glaucoma is considered to be a neurodegenerative disorder characterized by retinal ganglion cell (RGC) apoptosis. Endogenous CDP-choline is a natural precursor of phosphatydylcholine (PtdCho) synthesis. Enhancement of PtdCho synthesis has been hypothesized to inhibit neuronal apoptosis. Cytidine and choline are believed to individually cross neuronal cell membranes and facilitate PtdCho synthesis. A similar effect may be expected to occur in glaucomatous RGCs. Additionally, citicoline stimulates the dopaminergic system, which accounts for the major transmission of neural signals within the retina and precortical visual pathway [346,347,348].

### 3.22. Quercetin

Quercetin in an abundant flavonoid. It has strong antioxidative, anti-inflammatory, and neuroprotective functions in the body. It also has an anti-neoplastic and immunomodulatory agent. The literature has shown that is has been considered in multiple ocular conditions [349,350,351]. The antioxidative action by this substance is believed to be due to its donation of hydrogen ions to stabilize free radicals [352]. It also possesses an enhanced radical-scavenging activity [353]. Quercetin has also been shown to block the production of vascular endothelial growth factor in cancer models [354].

IOP is the most important modifiable factor in glaucoma management. Failure to properly manage the condition may result in various deficits, including loss of visual field [355,356]. Nutrients, including quercetin, have been shown to help in the management of glaucoma via several pathways in addition to the established methods of reducing IOP, protecting retinal optic ganglion cells, and improving perfusion rates to the optic nerve head [357,358,359]. Quercetin has been shown to induce neuroprotection in ganglion cells in studies involving excitotoxic destruction in rat glaucoma models [360]. The pathophysiology of the glaucomas includes retinal ganglion cell apoptosis and an increased photopic negative response. Quercetin has shown a photopic negative response, an index of glaucomatous damage in murine models, proving it to be a viable anti-glaucoma supplement [361]. Quercetin has also been shown to reduce apoptotic cell death, thereby protecting retinal ganglion cells in rat models [362,363].

### 3.23. Eyebright

Eyebright (*Euphrasia officinalis*) is a plant that has been utilized in contemporary eye care medication for decades [364]. Three extracts have been isolated from the leaves of this plant, namely ethanol, ethyl acetate, and heptane [365]. Studies have shown that euphrasia successfully reduced glycemic levels to baseline in rats, which had been previously induced using alloxan [366]. Blazics et al. were also able to show that euphrasia extracts had significant antioxidant properties [367]. Stoss et al. reported that eyebright eyedrop formulations were efficient in ameliorating conjunctivitis indices in a human population, as reported by [368].

Numerous studies have shown that the combination of factors and nutrients that function in synergy can provide an enhanced clinical outcome. Dolgova et al. examined the effect of a nutrient complex (Focus Forte) on POAG patients and reported an improvement in retina function [369]. Another study also reported therapeutic effects of nutrient-based combination therapies on a mouse model with elevated IOP. The study concluded that retinal ganglion cell loss was significantly reduced, thereby preventing visual dysfunction; thus, the authors recommended the use of the test mixture of forskolin, homotaurine, spearmint, and B vitamins in glaucoma management [370].

## 4. Discussion

Our study assessed some of the current literature regarding the use of several nutrients in medicine and modern-day ophthalmology. Table 2 shows the molecular characteristics of the nutrients obtained from the PubChem database (https://pubchem.ncbi.nlm.nih.gov/, accessed on 7 April 2023). This minireview was based on the past decade of literature available in research, thus it is limited and potentially has bias considering that important and fundamental studies published prior to 2013 have not been mentioned in this paper.

The rise of these nutritional supplements in the literature presents a new dawn in the management of numerous diseases in medicine, including glaucoma and other ophthalmic pathologies. The inexpensive nature of nutrient-rich substances, the absence of significant side effects when administered, and their easy absorption render them an attractive inclusive option in medical therapy. Supplements surely cannot replace traditional medical and surgical treatment; however, they can provide synergic positive effects to enhance the efficacy of gold-standard treatments and can help stabilize the general health conditions of the individual.

In conclusion, nutrients, antioxidants, vitamins, organic compounds, and micronutrients can be useful as integrative IOP-independent strategies in the management of glaucoma and other ophthalmologic pathologies. However, additional research and possibly large multicenter clinical trials are continually needed to confirm the efficacy of supplements as coadjutant therapeutic options in treatment regimens. It must be stated that the first-line managements for glaucoma continue to be therapeutic agents in the form of eye drops such as prostaglandin analogs, beta blockers, etc. [371]. Some of these drops have also shown inhibitory properties against the SARS-CoV-2 virus in situ [372]. Systemic medication is also recommended when topical medications alone cannot achieve the desired reduction in IOP.

**Table 2 life-13-01120-t002:** Molecular characteristics and sources of the reviewed nutrients (using PubChem at: https://pubchem.ncbi.nlm.nih.gov/, accessed on 7 April 2023).

Name	IUPAC	Sources	Chemical Structure
Glutathione	(2S)-2-amino-5-[[(2R)-1-(carboxymethylamino)-1-oxo-3-sulfanylpropan-2-yl]amino]-5-oxopentanoic acid	Avocados, okra, spinach	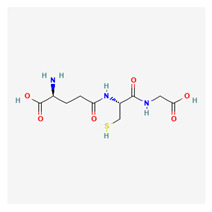 [373]
Minocycline	(2E,4S,4aR,5aS,12aR)-2-(Amino-hydroxy-methylidene)-4,7-bis(dimethylamino)-10,11,12a-trihydroxy-4a,5,5a,6-tetrahydro-4H-tetracene-1,3,12-trione	Tetracycline antibiotic	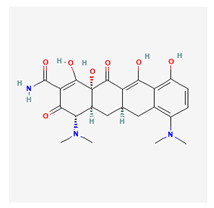 [374]
Spermidine	N-(3-Aminopropyl)butane-1,4-diamine	Soy protein, legumes, grain	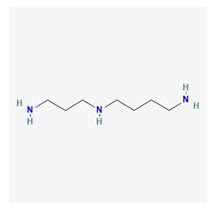 [375]
Fisetin	2-(3,4-Dihydroxyphenyl)-3,7-dihydroxychromen-4-one	Apples, onions, cucumbers	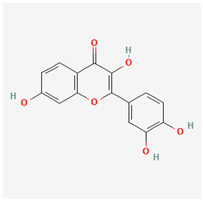 [376]
Omega-3s	(4Z,7Z,10Z,13Z,16Z,19Z)-docosa-4,7,10,13,16,19-hexaenoic acid;(5Z,8Z,11Z,14Z,17Z)-icosa-5,8,11,14,17-pentaenoic acid;(9Z,12Z,15Z)-octadeca-9,12,15-trienoic acid	Cod liver oil, salmon, mackerel, flaxseed	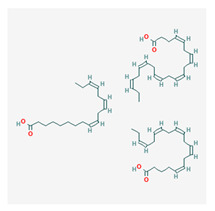 [377]
Rapamycin	(1R,9S,12S,15R,16E,18R,19R,21R,23S,24E,26E,28E,30S,32S,35R)-1,18-dihydroxy-12-[(2R)-1-[(1S,3R,4R)-4-hydroxy-3-methoxycyclohexyl]propan-2-yl]-19,30-dimethoxy-15,17,21,23,29,35-hexamethyl-11,36-dioxa-4-azatricyclo[30.3.1.04,9]hexatriaconta-16,24,26,28-tetraene-2,3,10,14,20-pentone	Macrolide compound	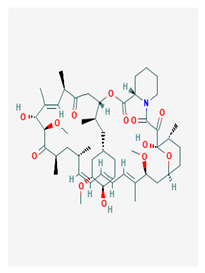 [378]
Metformin	3-(diaminomethylidene)-1,1-dimethylguanidine	French lilac	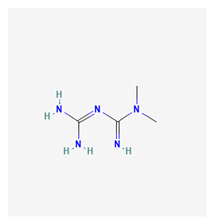 [379]
Alpha ketoglutarate	2-Oxopentanedioic acid	Metabolite	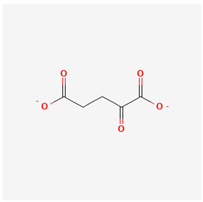 [380]
Vitamin B3 (niacin)	pyridine-3-carboxylic acid	Meats, nuts, legumes, bananas	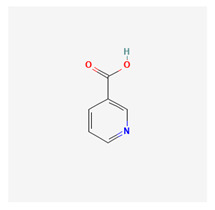 [381]
Vitamin D	(3β,5Z,7E,22E)-9,10-secoergosta-5,7,10(19),22-tetraen-3-ol	Sardines, tuna fish	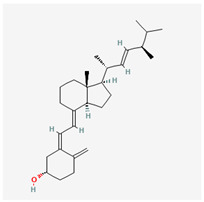 [382]
Lutein	β,ε-carotene-3,3’-diol	Green leafy vegetables such as kale and spinach	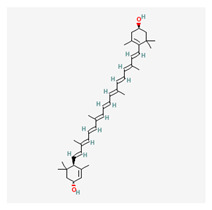 [383]
Zeaxanthin	β,β-carotene-3,3′-diol	Green leafy vegetables such as broccoli and spinach	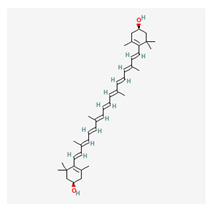 [384]
Resveratrol	3,5,4′-Trihydroxystilbene	Grapes and berries	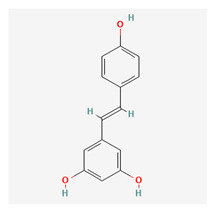 [385]
Pyruvate	2-oxopropanoic acid	Metabolite	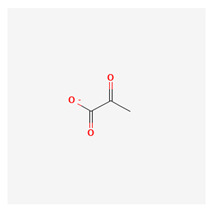 [386]
Vitamin A (retinol)	(2E,4E,6E,8E)-3,7-dimethyl-9-(2,6,6-trimethylcyclohexen-1-yl)nona-2,4,6,8-tetraen-1-ol	Liver, eggs	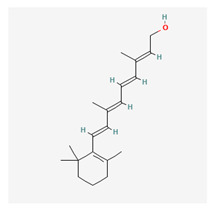 [387]
Vitamin B1 (thiamin)	2-[3-[(4-amino-2-methylpyrimidin-5-yl)methyl]-4-methyl-1,3-thiazol-3-ium-5-yl]ethanol	Wholegrain, fish	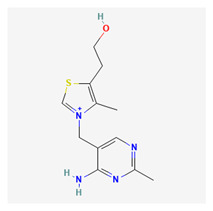 [388]
Vitamin B2 (riboflavin)	7,8-dimethyl-10-[(2S,3S,4R)-2,3,4,5-tetrahydroxypentyl]benzo[g]pteridine-2,4-dione	Dairy, cheese, lean beef	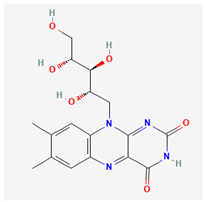 [389]
Vitamin B6 (folate)	(2*S*)-2-[[4-[(2-amino-4-oxo-3*H*-pteridin-6-yl)methylamino]benzoyl]amino]pentanedioic acid	Green leafy vegetables, fortified cereals	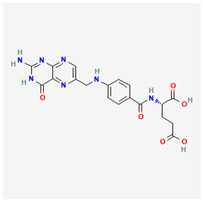 [390]
Vitamin B12 (cobalamin)	cobalt(3+);[(2*R*,3*S*,4*R*,5*S*)-5-(5,6-dimethylbenzimidazol-1-yl)-4-hydroxy-2-(hydroxymethyl)oxolan-3-yl] [(2*R*)-1-[3-[(1*R*,2*R*,3*R*,4*Z*,7*S*,9*Z*,12*S*,13*S*,14*Z*,17*S*,18*S*,19*R*)-2,13,18-tris(2-amino-2-oxoethyl)-7,12,17-tris(3-amino-3-oxopropyl)-3,5,8,8,13,15,18,19-octamethyl-2,7,12,17-tetrahydro-1*H*-corrin-21-id-3-yl]propanoylamino]propan-2-yl] phosphate;cyanide	Seafood and dairy products	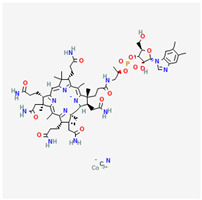 [391]
Vitamin C (ascorbate)	(2R)-2-[(1S)-1,2-dihydroxyethyl]-3,4-dihydroxy-2H-furan-5-one	Citrus fruits	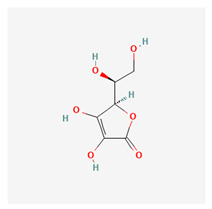 [392]
Vitamin E	(2R)-2,5,7,8-tetramethyl-2-[(4R,8R)-4,8,12-trimethyltridecyl]-3,4-dihydrochromen-6-ol	Wheat germ oil; sunflower, safflower, and soybean oil; sunflower seeds, almonds; peanuts, peanut butter.	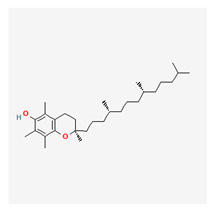 [393]
Citicoline	[[(2R,3S,4R,5R)-5-(4-amino-2-oxopyrimidin-1-yl)-3,4-dihydroxyoxolan-2-yl]methoxy-hydroxyphosphoryl] 2-(trimethylazaniumyl)ethyl phosphate	Organ meats; cauliflower, broccoli	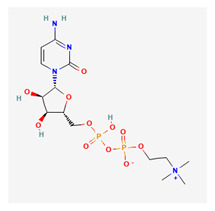 [394]
Quercetin	2-(3,4-dihydroxyphenyl)-3,5,7-trihydroxychromen-4-one	Citrus, red wine, apples, onions, sage, parsley	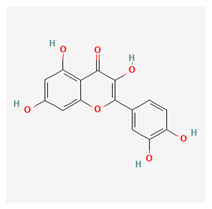 [395]

## Figures and Tables

**Figure 1 life-13-01120-f001:**
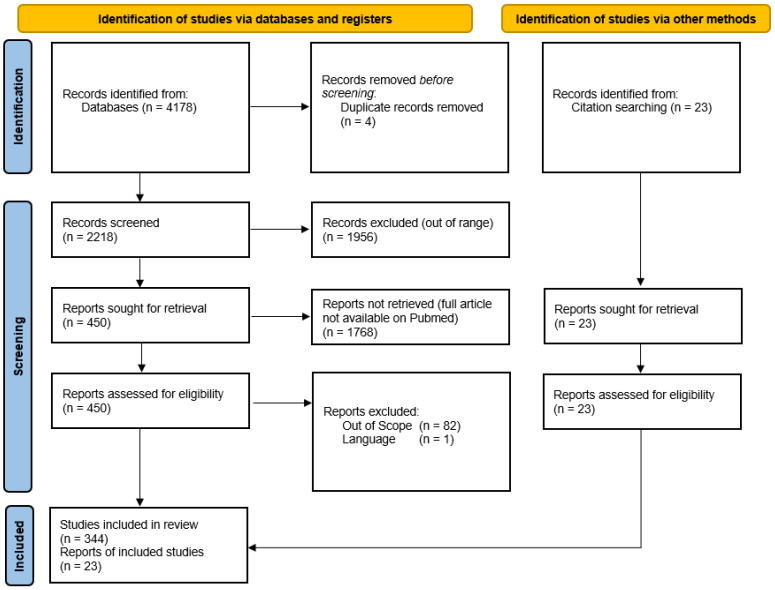
PRISMA flow diagram of the included studies, adapted from Ref. [25].

**Table 1 life-13-01120-t001:** Nutrients used in ophthalmology.

Nutrient	Methodology	Subjects	Number of Participants	Result	Reference
Glutathione	Case-control	Human	49 (19NTG, 30 normals)	Glutathione levels were statistically the same in the two groups	[26]
Gluthathione	Case-control	Human	77 (56 OAG, 21 normals)	A lower redox index was seen in the AOG group as compared with the normals	[27]
Gluthathione	Case-control	Human	113 (34 OAG, 30 NTG, 53 normals)	POAG and NTG group had significantly lower GSH levels	[28]
Minocycline	Experimental/Cohort	Rodent	159 rats	Minocycline 22 mg/kg upregulated survival genes in rat retina	[29]
Omega-3	Randomized control trial	Human	105 adults (all without diagnosis of glaucoma)	IOP reduced after 3 months of oral omega-3 supplementation in normotensive patients	[30]
Rapamycin	Experimental	Murine	Unspecified	Decreased intraocular pressure via aqueous ROCK inhibition following topical application	[31]
Metformin	Experimental	Mice	44 mice	Parenteral administration of rapamycin diminished free radicals in the trabecular meshwork	[32]
Niacin	Prospective	Human	50 patients with CRVO and macula edema	Niacin supplementation may play a vital role in improving structural and anatomical function in eyes with CRVO.	[33]
Zeaxanthin	Experimental	Murine	7 mice	Zeaxanthin supplementation protected RPE cells from mitochondrial oxidative stress	[34]
Lutein	Experimental	Human	99 volunteers	Lutein supplementation resulted in a significant improvement in macular pigment optical density.	[35]
Spermidine	Experimental	Human	17 patients with high glycemic variability (GV), 16 patients with low GV, and 21 normal patients for validation	Spermidine was significantly positively associated with glucose coefficient of variation (CV); hence, it plays some part in glucose regulation	[36]
AKG	Experimental	Human	4	AKG protected retinal pigment epithelium cells by rescuing mitochondrial function	[37]
Fisetin	Experimental	In vitro Human RPE Cells	-	Fisetin and luteolin protected ARPE-19 cells from oxidative stress-induced cell death	[38]
Vitamin C	Observational	Human in vivo tissue	138 individuals split into 69 normals and 69 individuals with established Vitamin C deficiency	There was a statistically reduced thickness of the retina and choroid in the group with Vitamin C deficiency as compared with the normals	[39]
Vitamin A, C, and E	Observational	Human in vivo ocular tissue	18,669 individuals who participated in the SUN project	A combination of these three nutrients was associated with a reduced risk of glaucoma	[40]
Citicoline, Homotaurine, Vitamin E	Experimental	Human	109 subjects	Contrast sensitivity scores and quality of life indices improved with nutrient supplementation	[41]

NTG = normal tension glaucoma, OAG = open-angle glaucoma, ARPE = adult retinal pigment epithelium, CRVO = central retinal vein occlusion.

## Data Availability

Data are available in a publicly accessible repository.

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
