# Peer review of "Nutritional Factors: Benefits in Glaucoma and Ophthalmologic Pathologies"

_life, 2023, doi:10.3390/life13051120_

Round 1

Reviewer 1 Report

In their manuscript entitled "The nutritional factor: Benefits in glaucoma and ophthalmologic pathologies. It’s not all about pressure" the authors review very interesting literature. The literature search was performed well.

However I am missing some important points:

- A conlcusion is missing - what is the intention - hypotheses the authors want to express

- A statistacal analysis and/or comparision of the individual cited studies.
so, a statistical anlaysis of the studies (e.g. meta analysis) with appropriate statistical method, to proof the significance of the Hypotheses would be very helpful.

In principle, the authors did already a good search, however to make the manuscript valubale for a high impact factor journal like "LIFE" amd to be recognized especially in the medical field.

This could be a statistical analyisis to proof the significance of the hypotheses or central "message".

Author Response

Reviewer 1

In their manuscript entitled "The nutritional factor: Benefits in glaucoma and ophthalmologic pathologies. It’s not all about pressure" the authors review very interesting literature. The literature search was performed well. 

Many thanks for the positive remarks about our manuscript.  

However, I am missing some important points:- A conclusion is missing - what is the intention - hypotheses the authors want to express 

In accordance with the Reviewer’s comments, a conclusive statement to summarize the aim of the review has been added to the Abstract and the Discussion sections stating that:

“Nutritional supplementation can thus be useful as integrative IOP-independent strategies in the management of glaucoma and of other ophthalmologic pathologies.” 

- A statistical analysis and/or comparison of the individual cited studies. so, a statistical analysis of the studies (e.g. meta-analysis) with appropriate statistical method, to proof the significance of the Hypotheses would be very helpful. In principle, the authors did already a good search, however to make the manuscript valuable for a high impact factor journal like "LIFE" and to be recognized especially in the medical field. This could be a statistical analysis to proof the significance of the hypotheses or central "message". 

In accordance with the comments made by the Reviewer, the description of the methodology and application of PRISMA guidelines have been rewritten to read: 

“Each study was independently assessed by at least two reviewers (M.M., G.A., E.E., E.T.,  and M.Z), and rating decisions were based on the consensus of the reviewing authors. A total of 374 references were included in the Review, as indicated in the Prisma in Table 1. This search strategy was limiting in the midst of the vast literature, which could have thus potentially and unintentionally excluded opinion leaders in this field of research.”

Reviewer 2 Report

This minireview discusses the potential benefits of using nutritional and integrative strategies, including nutrients, antioxidants, vitamins, organic compounds, and micronutrients, as IOP-independent approaches to delay or halt the progression of glaucomatous retinal ganglion cell degeneration. The review examines the molecular and biological characteristics, neuroprotective activities, antioxidant properties, and beneficial mechanisms of various substances proposed in the management of ophthalmologic diseases, particularly glaucoma, and highlights the need for large multicenter clinical trials to confirm these potential benefits. Such trials could pave the way for alternative and/or coadjutant therapeutic options in the management of glaucoma and other ocular pathologies. 

Introduction 

  1. Lack of specific references: The text makes many statements without citing specific studies or sources. More specific references to the literature would improve the credibility of the text. 

  1. Insufficient evidence for claims: The text suggests that nutritional supplements can lead to reduced rates of disease progression and a better quality of life for glaucoma patients, but there is little evidence cited to support these claims. More research is needed to fully understand the potential benefits of nutritional supplements in glaucoma management. 

  1. Limited scope of the study: The text states that the authors do not intend to provide an exhaustive analysis, but a quick overview of the use of nutrients in glaucoma and other ophthalmic conditions. This disclaimer may limit the overall impact and credibility of the study. 

The text describes a study that assessed the current literature regarding the use of several nutrients in medicine and ophthalmology. However, the study's methodology is not clearly described, as it only provides a brief overview of the search query and inclusion/exclusion criteria. Additionally, the text mentions the use of the PRISMA guidelines for systematic reviews, but only provides a figure without describing how these guidelines were applied. A more detailed description of the study's methodology and application of PRISMA guidelines would improve the credibility of the study. Furthermore, the text only searched the Pubmed database and limited the inclusion criteria to articles published in the last decade, which may lead to a bias towards more recent research and potentially missing relevant studies.

Author Response

Reviewer 2

This minireview discusses the potential benefits of using nutritional and integrative strategies, including nutrients, antioxidants, vitamins, organic compounds, and micronutrients, as IOP-independent approaches to delay or halt the progression of glaucomatous retinal ganglion cell degeneration. The review examines the molecular and biological characteristics, neuroprotective activities, antioxidant properties, and beneficial mechanisms of various substances proposed in the management of ophthalmologic diseases, particularly glaucoma, and highlights the need for large multicenter clinical trials to confirm these potential benefits. Such trials could pave the way for alternative and/or coadjutant therapeutic options in the management of glaucoma and other ocular pathologies. 

We thank the Reviewer for a very thorough and meticulous evaluation of our manuscript. Excellent points have been raised that can potentially improve our manuscript. We have tried to address each point, which has been listed below each comment.  

Introduction: Lack of specific references: The text makes many statements without citing specific studies or sources. More specific references to the literature would improve the credibility of the text.  

Specific references have been added to statements lacking them throughout the text, as requested.  

Insufficient evidence for claims: The text suggests that nutritional supplements can lead to reduced rates of disease progression and a better quality of life for glaucoma patients, but there is little evidence cited to support these claims. More research is needed to fully understand the potential benefits of nutritional supplements in glaucoma management.  

  • The concepts regarding the therapeutic benefits of the various nutritional factors have been toned down to avoid giving the impression that supplements can replace traditional therapy in ophthalmology or medicine.
  • The Reviewer makes a good point regarding the issue of enhanced quality of life and reduced progression, which could be misleading. Mention regarding these points has been deleted accordingly throughout the text.
  • In accordance with the comments made by the Reviewer, a sentence in the conclusion paragraph has also been reworded to read … “Additional research and possibly large multicenter clinical trials, however, are continually needed to confirm the efficacy of supplements as coadjutant therapeutic options in treatment regimens..”.

Limited scope of the study: The text states that the authors do not intend to provide an exhaustive analysis, but a quick overview of the use of nutrients in glaucoma and other ophthalmic conditions. This disclaimer may limit the overall impact and credibility of the study.  

This statement has been deleted from the Introduction to improve the credibility of the review as nicely pointed out by the Reviewer.  

The text describes a study that assessed the current literature regarding the use of several nutrients in medicine and ophthalmology. However, the study's methodology is not clearly described, as it only provides a brief overview of the search query and inclusion/exclusion criteria. Additionally, the text mentions the use of the PRISMA guidelines for systematic reviews, but only provides a figure without describing how these guidelines were applied. A more detailed description of the study's methodology and application of PRISMA guidelines would improve the credibility of the study.  

In accordance with the comments made by the Reviewer, the description of the methodology and application of PRISMA guidelines have been rewritten to read: 

“Each study was independently assessed by at least two reviewers (M.M., G.A., E.E., E.T.,  and M.Z), and rating decisions were based on the consensus of the reviewing authors. A total of 374 references were included in the Review, as indicated in the Prisma in Table 1. This search strategy was limiting in the midst of the vast literature, which could have thus potentially and unintentionally excluded opinion leaders in this field of research.”

 Furthermore, the text only searched the PubMed database and limited the inclusion criteria to articles published in the last decade, which may lead to a bias towards more recent research and potentially missing relevant studies.

 With regard to the limitations of the study, considering the vast amount of literature available in this field, we decided to assess the last decade of research and use this point as an inclusion criterion. The following has been added to the Discussion section to report this important limit that the Reviewer has rightfully stated:

“This minireview was based on the past decade of literature available in research, thus is limiting and potentially with bias considering that important and fundamental studies published prior to 2013 have not been mentioned in this paper.”

Your valuable comments and assistance with our paper are greatly appreciated. We look forward to your final decision regarding our modifications and are ready to address further issues if necessary. We hope that all concerns have been addressed in an appropriate manner.

Reviewer 3 Report

The review is a description of the importance of nutrients and other compounds as therapeutic option for glaucoma. It is a too long description, sometimes tedious, that authors should shorten. Also title should be changed as "Nutritional factors: benefits in glaucoma and ophthalmologic pathologies".

Minor comments:

- line 36: remove comma and put dot.

- Simply summary, materials and methods and results are not in guidelines of journal for a review article.

- Table 1 and 2 are so large and long. Please reduce them.

- You should reduce all nutrients description but emphasize other therapies, such as eye drops. Please see the following articles: 

Gazzard G, Konstantakopoulou E, Garway-Heath D, Adeleke M, Vickerstaff V, Ambler G, Hunter R, Bunce C, Nathwani N, Barton K; LiGHT Trial Study Group. Laser in Glaucoma and Ocular Hypertension (LiGHT) Trial: Six-Year Results of Primary Selective Laser Trabeculoplasty versus Eye Drops for the Treatment of Glaucoma and Ocular Hypertension. Ophthalmology. 2023 Feb;130(2):139-151; 

Petrillo F, Chianese A, De Bernardo M, Zannella C, Galdiero M, Reibaldi M, Avitabile T, Boccia G, Galdiero M, Rosa N, Franci G. Inhibitory Effect of Ophthalmic Solutions against SARS-CoV-2: A Preventive Action to Block the Viral Transmission? Microorganisms. 2021 Jul 21;9(8):1550. 

Author Response

Reviewer 3

The review is a description of the importance of nutrients and other compounds as therapeutic option for glaucoma. It is a too long description, sometimes tedious, that authors should shorten. Also, title should be changed as "Nutritional factors: benefits in glaucoma and ophthalmologic pathologies". 

In accordance with the suggestions made by the Reviewer, the text has been slightly shortened. The title has been changed to “Nutritional factors: benefits in glaucoma and ophthalmologic pathologies”.

 Minor comments:

- line 36: remove comma and put dot.

- Simply summary, materials and methods and results are not in guidelines of journal for a review article.

- Table 1 and 2 are so large and long. Please reduce them.

The slight modifications listed by the Reviewer have been made throughout the text accordingly. The row of “location of research” has been deleted from Table 1. The row of “chemical formula” has also been deleted from the table.

- You should reduce all nutrients description but emphasize other therapies, such as eye drops. Please see the following articles:

Gazzard G, Konstantakopoulou E, Garway-Heath D, Adeleke M, Vickerstaff V, Ambler G, Hunter R, Bunce C, Nathwani N, Barton K; LiGHT Trial Study Group. Laser in Glaucoma and Ocular Hypertension (LiGHT) Trial: Six-Year Results of Primary Selective Laser Trabeculoplasty versus Eye Drops for the Treatment of Glaucoma and Ocular Hypertension. Ophthalmology. 2023 Feb;130(2):139-151;

Petrillo F, Chianese A, De Bernardo M, Zannella C, Galdiero M, Reibaldi M, Avitabile T, Boccia G, Galdiero M, Rosa N, Franci G. Inhibitory Effect of Ophthalmic Solutions against SARS-CoV-2: A Preventive Action to Block the Viral Transmission? Microorganisms. 2021 Jul 21;9(8):1550.

Impertinent statements regarding nutrient descriptions have been deleted throughout the text. In accordance with the suggestion made by the Reviewer, the following has been added about eye drop therapy based on the citations listed:

It must be stated that the first line management for glaucoma remains therapeutic agents in the form of eye drops like prostaglandin analogs, beta blockers, etc. [387]. Some of these drops have also shown inhibitory properties against the Sars-Cov-2 virus in situ [388]. Systemic medication is also recommended when topical medications alone cannot achieve the desired reduction in IOP.

Your valuable comments and assistance with our paper are greatly appreciated. We look forward to your final decision regarding our modifications and are ready to address further issues if necessary. We hope that all concerns have been addressed in an appropriate manner.  

Reviewer 4 Report

Manuscript Life-2298809

The nutritional factor: Benefits in glaucoma and ophthalmologic pathologies. It’s not all about pressure” for Life

Comments:

1.      In the paragraph on rapamycin, the authors mentioned its immunosuppressive activity. Please briefly elaborate on what immune mechanisms associated with immunosuppression are activated in this case.

2.      Similarly in the description of Zeaxanthin. Line 557. What does 'and immune system capabilities' mean? At the moment it is not very informative. Please expand on this issue.

3.      Please also mention in a separate chapter about mixtures of substances, such as plant extracts. In this context, please also mention the eyebright (Euphrasia) and its potential role in limiting glaucoma. Individual substances must act in a specific cellular context, interacting with other factors present in the tissue. The interrelationships between these factors are also worth mentioning. Is it possible to obtain a synergistic effect as a result of them?

Author Response

Reviewer 4

“The nutritional factor: Benefits in glaucoma and ophthalmologic pathologies. It’s not all about pressure” for Life.

Comments: 1. In the paragraph on rapamycin, the authors mentioned its immunosuppressive activity. Please briefly elaborate on what immune mechanisms associated with immunosuppression are activated in this case. 

With regards to the immunosuppressive activity of rapamycin, the following has been added about the immune mechanisms associated with immunosuppression to read:

Inhibition of mTOR by rapamycin suppresses the immune response by preventing cell cycle progression from G1 to S phase, thereby blocking proliferation [117].” 

  1. Similarly in the description of Zeaxanthin. Line 557. What does 'and immune system capabilities' mean? At the moment it is not very informative. Please expand on this issue.

With regard to immune system capabilities, this issue has been toned down in this paragraph. The text has been modified to briefly mention literature in this field as follows:

“A study in healthy elderly men found that supplementation with lutein and zeaxanthin improved immune function by increasing the activity of natural killer cells [166].” 

  1. Please also mention in a separate chapter about mixtures of substances, such as plant extracts. In this context, please also mention the eyebright (Euphrasia) and its potential role in limiting glaucoma. Individual substances must act in a specific cellular context, interacting with other factors present in the tissue. The interrelationships between these factors are also worth mentioning. Is it possible to obtain a synergistic effect as a result of them?

The Reviewer makes a valid observation. In accordance with the suggestion, several paragraphs with appropriate references have been added to create a new chapter about eyebright before the Discussion Section. Mention regarding the interrelationship between factors has been included to read:  

Eyebright (Euphrasia officinalis) is a plant that has been utilized in contemporary eye care medication for decades [387]. Three extracts have been isolated from the leaves of this plant namely; ethanol, ethyl acetate, and heptane [388]. Studies have shown that euphrasia successfully reduced glycemic levels to baseline in rats, which had been previously induced using Alloxan [389]. Blazics et al. were also able to show that euphrasia extracts had significant antioxidant properties [390]. Stoss et al. reported that eyebright eyedrop formulations were efficient in ameliorating conjunctivitis indices in a human population as reported by [391].

     Numerous studies have shown that the combination of factors and nutrients that function in synergy can provide an enhanced clinical outcome. Dolgova et al. examined the effect of a nutrient complex (Focus Forte) on POAG patients and reported an improvement in retina function [392]. Another study also reported therapeutic effects of nutrient based combination therapies on a mouse model with elevated IOP. The study concluded that retinal ganglion cell loss was significantly reduced, thereby preventing visual dysfunction, and thus recommended the use of the test mixture of Forskolin, Homotaurine, Spearmint, and B Vitamins in glaucoma management [393].” 

Your valuable comments and assistance with our paper are greatly appreciated. We look forward to your final decision regarding our modifications and are ready to address further issues if necessary. We hope that all concerns have been addressed in an appropriate manner.

Round 2

Reviewer 1 Report

The authors revised the manuscript very well and added significant points.
I recommend for publiction

Reviewer 2 Report

Thank you for submitting the revised manuscript, which we have carefully evaluated. Although we appreciate the efforts made to address the issues raised in the first review, we regret to inform you that the revised manuscript still does not meet the high scientific standards required for publication in our journal.

Although you have added specific references to statements lacking them throughout the text, the manuscript still lacks a clear and concise description of the methodology used for the literature search and the application of PRISMA guidelines. Additionally, the search strategy has limitations, as it only searched the PubMed database and included articles published in the last decade, which may lead to a biased view of the literature and potentially missing relevant studies.

Furthermore, although the revised manuscript toned down the language regarding the therapeutic benefits of the various nutritional factors, it still lacks a comprehensive discussion of the limitations and potential adverse effects of using nutritional supplements in the management of glaucoma and other ophthalmic conditions.

Reviewer 3 Report

The authors improved the quality of the manuscript that is now suitable for publication.